# Translation, Adaptation and Assessment of the Psychometric Properties of the Mental Health Knowledge Questionnaire in a Sample of Higher Education Students in Portugal

**DOI:** 10.3390/ijerph20043022

**Published:** 2023-02-09

**Authors:** Cláudia Chaves, João Duarte, Francisco Sampaio, Joana Coelho, Carlos Sequeira

**Affiliations:** 1Superior School of Health, Polytechnic of Viseu, Rua Dom João Crisóstomo Gomes de Almeida, 102, 3500-843 Viseu, Portugal; 2Nursing School of Porto, Rua Dr. António Bernardino de Almeida, 830, 844, 856, 4200-072 Porto, Portugal; 3CINTESIS@RISE, Nursing School of Porto (ESEP), Rua Dr. Plácido da Costa, 4200-450 Porto, Portugal; 4Northern School of Health of the Portuguese Red Cross, Rua da Cruz Vermelha Cidacos-Apartado 1002, 3720-126 Oliveira de Azeméis, Portugal

**Keywords:** mental health, knowledge, health literacy, students, psychometrics, mental disorders

## Abstract

A significant part of the worldwide population is affected by some mental disorder. Previous research conducted with the general population has revealed poor knowledge when it comes to mental health. Therefore, it is imperative to assess mental health literacy using robust assessment tools. Thus, this study aimed to translate, adapt and assess the psychometric properties of the Mental Health Knowledge Questionnaire in a sample of higher education students in Portugal. This study used a sample consisting of 2887 participants. Internal consistency for the psychometric study was estimated using Cronbach’s alpha coefficient. Construct validity was tested using exploratory and confirmatory factor analysis, convergent validity, and discriminant validity. After data analysis, the final Portuguese version of the Mental Health Knowledge Questionnaire was composed of 14 items. The goodness-of-fit indices were adequate, confirming the quality of the model’s fit to the empirical data (χ²/df = 2.459, GFI = 0.983, CFI = 0.969, RMSEA = 0.032, RMR = 0.023, SRMR = 0.032). This assessment tool is valid and reliable to assess higher education students’ mental health literacy in Portugal. Analyses to confirm the scale’s external validity, measurement equivalence and replicability are still required.

## 1. Introduction

The definition of mental health literacy has evolved over the last few years. The concept was first introduced by Jorm et al. [1], who defined it as “knowledge and beliefs about mental disorders which aid their recognition, management or prevention”. More recently, Kutcher et al. [2] gave a broader definition of mental health literacy as “understanding how to obtain and maintain positive mental health; understanding mental disorders and their treatments; decreasing stigma related to mental disorders; and enhancing help-seeking efficacy (knowing when and where to seek help and developing competencies designed to improve one’s mental health care and self-management capabilities)”.

Understanding the importance of health literacy should be a cross-cutting issue for every generation. However, this awareness should be an even greater concern when dealing with younger age groups as they represent the next generations of adults. Data from the WHO World Mental Health International College Student Initiative [3] indicated that only 24.6% of first-year students would seek help in case of an emotional problem. Furthermore, only 16.4% of them received healthcare, despite the high prevalence of mental health disorders and their well-documented consequences [4]. That way, the focus may be placed on new generations, such as higher education students, since there are common circumstances experienced by most of these students that can negatively affect their health [5].

Concurrently, there has been much debate among the scientific community in recent years, on the undeniable importance of mental health literacy and on a new and promising concept called “positive mental health literacy” [6]. The positive component of mental health literacy refers to a person’s awareness of how to achieve and maintain good mental health [7]. These two related concepts have proven critical health promotion, even among higher education students. This proactive behaviour is essential to achieve stronger free will, self-knowledge, and self-esteem and to enable a person to take charge of everything that may influence their life [8].

The literature argues that achieving a good level of health literacy is a determining factor in the conduct of every citizen since it will enable them to gain control over their own life [8]. The literature also argues that investigating the levels of mental health literacy among higher education students will contribute to fighting the stigma towards mental disorders, demystifying false beliefs and stereotypes, and providing a new inclusive approach to every member of society [9].

In Portugal, in a study carried out in 2013 [10], around a quarter of 4938 young Portuguese people aged 14–24 years failed to recognize instances of depression. In the same sample, only 42.17% and 22.21% of the participants were able to recognize schizophrenia and psychosis, respectively [11]. However, both studies focused on recognizing mental disorders and not on the knowledge about positive mental health, i.e., the capacity to feel, think and act in ways that enhance the ability to enjoy life and deal with the challenges one faces [12].

Most assessment tools on mental health literacy usually focus on knowledge about mental disorders. For example, the Mental Health Literacy Scale [13], which was validated for the Portuguese population in 2021 [14], is entirely focused, for instance, on the recognition of mental disorders, knowledge of their risk factors and causes, knowledge of self-treatments, or knowledge of available professional help. However, the assessment tool Mental Health Knowledge Questionnaire (MHKQ) does not concretely focus on mental disorders. Even the items which are included in the dimension “Knowledge of the characteristics of mental health and mental disorders” (e.g., “Positive attitudes, good interpersonal relationships and a healthy lifestyle can help maintain mental health”) are not focused on signs and/or symptoms of mental disorders, but on knowledge about how to maintain well-being. 

The MHKQ was developed by the Chinese Ministry of Health in 2009 to evaluate public knowledge and mental health awareness [15]. In its original version, the scale was composed of 20 statements; items 1 to 16 were about mental health, and items 17 to 20 asked whether respondents had heard of four mental health promotion days (e.g., International Mental Health Day or International Suicide Prevention Day) [16]. In the original questionnaire, each respondent was required to select “true”, “false”, or “unknown” about statements concerning mental health. Correct responses were “true” for items 1, 3, 5, 7, 8, 11, 12, 15 and 16 and “false” for items 2, 4, 6, 9, 10, 13 and 14. One point was given to each correct answer, and 0 points were given to incorrect or unknown answers. Items 17–20 were scored with 1 or 0 points to those answering “yes” and “no”, respectively [15].

A previous preliminary study using an exploratory and confirmatory factorial analysis was conducted with a sample of Portuguese postpartum women. The original instrument was translated into Portuguese and adapted to the Portuguese population using the same categorisation criteria. The outcomes showed a three-factor structure that included 14 items, which was not in line with the structure suggested by the original authors. Only the factor that included items 17, 18, 19 and 20 remained constant. In light of these outcomes, considering the non-objectivity of the construct (mental health literacy), and in order to make the scale more consistent, a Likert-type response format with five response options was adopted. The new version of the adapted scale was developed based on these recommendations. Thus, the study with a sample of postpartum women was used as a first approach to analyse how the scale would perform in a Portuguese sample, which led to a change in its response format that later would need to be tested in a different sample.

Therefore, despite that preliminary study, the MHKQ was not validated for the Portuguese population and, most specifically, for higher education students in Portugal. Therefore, to fill that gap, this study aimed to translate, adapt and validate the MHKQ in a sample of higher education students in Portugal.

## 2. Materials and Methods

### 2.1. Design, Participants and Setting

A psychometric study was carried out since this is the most commonly used approach to validate self-report instruments to measure latent constructs. This type of research uses quantitative procedures in order to obtain the validity, reliability and standardisation of the measurement instrument [17].

This study was based on a non-probability convenience sample comprising 2887 higher education students from several universities in mainland Portugal and the Islands. Most participants were single (90.9%) with a mean age of 23.13 years (±6.51). Most respondents attended undergraduate studies (90.8%) in higher education institutions located in the North and Centre of the country, and 72.1% attended healthcare courses.

The psychometric study was conducted with a sample randomly divided into two groups (each group consisted of approximately 50% of the participants) using the option “data, select case, random sample case” in IBM SPSS Statistics. One of the groups was used to perform exploratory factor analysis, while the other was used to perform confirmatory factor analysis.

Data collection took place between October 2019 and March 2020. The only inclusion criterion defined was that the participants had to be students attending a higher education institution who had freely accepted to participate in the study once the final informed consent was validated. All participants were provided with the necessary information and asked to complete and validate the informed consent form.

### 2.2. Instrument 

Past psychometric testing of the MHKQ has reported internal consistency of Cronbach’s alpha coefficients ranging from 0.57 to 0.73 and a 2-week test–retest reliability of 0.68, as measured by intraclass correlation coefficients [18]. The exploratory factor analysis performed by Yu et al. [16] yielded a three-factor solution: knowledge of the characteristics of mental health and mental disorders encompassing items 1, 2, 3, 5, 7, 8, 11, 12, 15 and 16, belief in the epidemiology of mental disorders, which included items 4, 6, 9, 10, 13 and 14, and awareness of mental health promotion activities, comprising items 17, 18, 19 and 20. Cronbach’s alpha coefficients ranged from 0.62 to 0.67.

### 2.3. Procedures

#### 2.3.1. Procedures Followed to Adapt the Mental Health Knowledge Questionnaire

The translation and adaptation of the MHKQ for European Portuguese were based on: (1) technical review and semantic analysis adapted to a Likert-type scale; (2) content validation; and (3) administration of the pretest to a sample of the target population to assess respondents’ level of understanding.

Thus, firstly, the existing dichotomous scale of the MHKQ items was converted into a Likert-type scale in which each item was given five different response options: (1) strongly disagree, (2) disagree, (3) neither agree nor disagree, (4) agree, and (5) strongly agree. Items 2, 4, 6, 9, 10, 13, 14 and 15 are reverse-scored. Items 17, 18, 19 and 20 are exceptions since they maintained their initial categorisation (“Yes” = 1 point; “No” = 0 points). In doing so, linguistic equivalence was ensured, an outcome that had already been achieved in the study conducted with postpartum women, as was conceptual and psychometric equivalence. Conceptual equivalence was obtained through expert consultation, and psychometric equivalence was established through the revalidation of the instrument and the analysis of the psychometric properties.

To successfully adapt the document to a new format (Likert-type scale), it was submitted to a review conducted by a panel of experts composed of an expert in psychometrics, a healthcare worker experienced in mental health situations, a nursing professor, the translators who translated and back-translated the assessment tool (MHKQ) and a member of the research team (C.C.), who was responsible for assessing the semantic and idiomatic equivalence, the conceptual equivalence, and the cultural equivalence of this new format.

Once the necessary changes were made, a pretest was administered to a sample of 50 higher education students to assess the respondents’ level of understanding. The qualitative assessment showed that the questions were clear and that there was a high level of agreement among the respondents. 

#### 2.3.2. Data Analysis Procedures

After being adapted to a Likert-type scale written in Portuguese, the MHKQ was validated using multivariate exploratory and confirmatory factor analysis techniques.

The exploratory factor analysis was conducted to uncover the underlying structure in a data matrix and determine the number and nature of the latent variables (factors) that best represent a set of observed variables [19,20]. This statistical technique allows a more accurate exploration of the underlying dimensions, constructs or latent variables of the observed variables.

On the other hand, confirmatory factor analysis is a procedure whose purpose is to analyse the relationships between a set of indicators or observed variables and one or more latent variables or factors. It is usually used when there is already previous information on the factorial structure that needs to be confirmed [20].

The psychometric properties of the MHKQ were assessed based on its reliability and validity, as these are the most appropriate procedures to ensure the informative quality of the data. Reliability is crucial in determining the internal consistency or homogeneity of the items, namely by determining the Pearson correlation coefficient between the different questions and the overall score and Cronbach’s alpha coefficient. To achieve a good item definition, some authors [20,21] only consider the items or variables with correlation coefficients (*r*) higher than 0.2 with the overall score when this score does not include that specific item. 

Regarding validity, only concept or construct validity and hypothetic-deductive validity [22], which provide information on the quality of the items, were considered. Construct validity involves a statistical approach that includes factor analysis of the items and the results. This method consists of an exploratory data analysis technique whose purpose is to discover and analyse the structure of a set of interrelated variables to build a measurement scale [20]. Several methods can be used to achieve that aim, but the “principal components analysis” is mostly adopted. The varimax orthogonal rotation was the factor rotation method selected.

To achieve a good factor definition, the items with loading levels (r) higher than or equal to 0.4 were considered. Eigenvalues higher than 1 (one) and the scree plot were considered to determine which factors should be retained. IBM SPSS Statistics version 25 for Windows was used for data processing. Additionally, through the Factor software, the analysis was implemented using a polychoric matrix and Robust Diagonally Weighted Least Squares (RDWLS) extraction method [23]. The decision on the number of factors to be retained was performed based on the Parallel Analysis technique with a random permutation of the observed data [24]. Model fit was assessed using the root mean square error of approximation (RMSEA), comparative fit index (CFI) and Tucker–Lewis Index (TLI) fit indices. Then, the other half of the sample was used in a confirmatory factor analysis to assess the goodness-of-fit of the measurement models of the subscales. Maximum likelihood estimation method was combined with the AMOS software version 25. Data analysis was conducted using several statistical procedures deemed to be the most adequate, namely: (i) item distribution, assessed using skewness (Sk) and kurtosis (Ku), and all items with absolute skewness values higher than 3 and kurtosis higher than 7 were removed; (ii) construct validity, assessed using factor validity, convergent validity and discriminant validity; and (iii) goodness-of-fit of the factor model, assessed based on the following indices and reference values: χ²/df (the ratio between chi-square and degrees of freedom)—the model’s fit is considered good if the ratio (χ²/df) is lower than 2, acceptable when it is lower than 5 and unacceptable when it is higher than 5; root mean square residual (RMR)—the lower it is, the better the model’s fit (e.g., RMR = 0 indicates a perfect fit); standardized root mean square residual (SRMR)—the standardized difference between the observed and the predicted correlation (a zero value indicates a perfect fit and a value lower than 0.08 is generally considered a good fit); goodness-of-fit index (GFI)—values higher or close to 0.95 are recommended, but any value above 0.90 is considered a good fit; comparative fit index (CFI)—is an additional fit index (values lower than 0.90 indicate a poor fit, values between 0.90 and 0.95 are evidence of a good fit, and values higher than 0.95 reflect a very good fit (this index can be a reference regardless of the size of the sample); root mean square error of approximation (RMSEA)—reference values between 0.05 and 0.08 represent a good fit and a very good fit occurs when the index is lower than 0.05; local goodness of fit assessed by factor loadings and by individual item reliability—reference values are 0.5 and 0.25, respectively, although in early studies factor loading values of 0.40 or close can be accepted. 

The model fit was performed drawing on the modification indices proposed by AMOS, and trajectories with modification indices above 11 were adjusted; construct reliability was assessed using composite reliability (CR), which indicates the degree to which the items are consistent manifestations of the latent factor (CR ≥ 0.7 indicates appropriate construct reliability).

The convergent validity of each factor was assessed using average variance extracted (AVE). Convergent validity was achieved when AVE was higher than 0.5 [18]. Factor discriminant validity was obtained by squaring the correlation between the factors. There is evidence of discriminant validity when the value obtained is lower than the AVE for each factor. 

All factors were standardized by setting their variances at 1.00 (one) and keeping them inter-correlated.

#### 2.3.3. Ethical Procedure

The study protocol was approved by the Ethics Committee of the Nursing School of Porto (protocol code: 2019/1945). All participants were provided with the necessary information and asked to sign the informed consent form. Anonymity, data confidentiality, and autonomy were ensured. Participants were also informed that they were free to participate in the study and could withdraw at any time. They were informed that they would not receive any compensation for participating and withdrawal from the study would not result in any harm/loss.

## 3. Results

### 3.1. Exploratory Factor Analysis

Table 1 presents the statistics (mean and standard deviation), item-total correlation, Cronbach’s alpha, and McDonalds’ omega coefficients of the 20 items of the scale based on the responses of the 1443 sample participants. The average indices in all items are higher than the expected average score, which means that the data are well-centred.

The Cronbach’s alpha coefficients of the items are barely reasonable, ranging between 0.661 (item 4—“All mental disorders are caused by external stressors”) and 0.710 (item 15—“Individuals with a bad temperament are more likely to have mental problems”), with an overall Cronbach’s alpha of 0.691. The value of the split-half reliability index is slightly lower than the overall alpha for both the first half (α = 0.563) and the second half (α = 0.469).

The McDonalds’ omega coefficients are more consistent, presenting the lowest value on item 11 (ω = 0.695) and the highest value on item 15 (ω = 0.740). The overall McDonalds’ omega coefficient is 0.726.

The factor analysis of the scale was conducted and included all the items. The Kaiser–Meyer–Olkin (KMO) test for sampling adequacy, which assesses the model’s adequacy, revealed a value of 0.809, suggesting the existence of a good correlation between the items. Bartlett’s Test of Sphericity (specificity test) was x^2^ = 4559.4980; *p* = 0.001, which indicates that the items are correlated. These test results support the use of exploratory factor analysis with these data. 

Factors were then extracted using principal component analysis and varimax orthogonal rotation, and eigenvalues greater than 1 and item loading levels higher than 0.40 were considered for factor retention [20]. A factorial structure comprising five factors, which explained 48.74% of the total variance, was then obtained. However, the scree plot showed a three-factor structure according to the inflection point of the curve (Figure 1). This finding, combined with two factors that only included two items each, led to a three-factor rotation. 

The results obtained with this new procedure led to the removal of items 6 and 9 because they showed loading scores lower than 0.40. There was a decrease in the explained variance (40.87%), while the KMO value increased (0.815). The anti-image matrix indicated the existence of high correlations ranging between *r* = 0.649 and *r* = 0.884.

Factor 1 ended up with eight items (1, 3, 5, 7, 8, 11, 12 and 16) that explained 18.53% of the total variance. This factor was referred to as the knowledge of mental health characteristics and mental disorders and presented an eigenvalue of 3.336.

Factor 2 comprised six items (2, 4, 10, 13, 14 and 15) that explained 11.79% of the total variance. It was called belief in the epidemiology of mental disorders and presented an eigenvalue of 2.212. 

Factor 3 included four items (17, 18, 19 and 20) that explained 10.55% of the total variance. It was dubbed awareness of health promotion activities and presented an eigenvalue of 1.900. Table 2 presents the loadings of the items per factor and reveals that all items had factor loadings higher than 0.40. Based on these results, a confirmatory factor analysis was then conducted.

The parallel analysis reinforced the existence of three factors since the percentages of the actual variance explained in the three factors were higher than those obtained by the resampling procedures. 

On the other hand, the goodness-of-fit indices obtained with the exploratory factor analysis indicate a quite adequate adjustment (RMSEA = 0.026; CFI = 0.982; GFI = 0.985).

### 3.2. Confirmatory Factor Analysis

The hypothesized three-factor model was submitted to confirmatory factor analysis using the maximum likelihood method. In assessing item sensitivity, we observed skewness and kurtosis values that suggested the existence of symmetry and normokurtosis in most of the items. Items 1 and 8 were the exceptions, as they were slightly skewed to the right but perfectly flat. In subsequent tests, these items were removed. Mardia’s coefficients of multivariate skewness and kurtosis (1.410) were lower than the reference value (5.0). Table 3 presents the critical ratios and the items’ loading values. Critical ratios showed that all items were statistically significant with the corresponding factor. Some items had very low loading values, so they were removed from subsequent analyses. 

Figure 2 represents the graphical output of the early model with the trajectories between the items and their respective factors, the factor loadings and the individual reliability of the items. The model presented an adequate overall goodness-of-fit for all indices (χ²/df = 3.117, GFI = 0.940, RMSEA = 0.058, RMR = 0.053, and SRMR = 0.056), except for CFI (=0.864).

The model was refined as the most problematic items were removed, and the trajectories were adjusted using the modification indices suggested by AMOS. The results are presented in Figure 3. In factor 1, items 1 and 8 were removed, and so were items 2 and 15 in factor 2. The modification indexes highlight the association between errors 9 and 12, included in factor 2, and errors 16 and 18 in factor 3. The overall fit indices revealed values that are appropriate for all indices under analysis: χ²/df = 2.459, GFI = 0.983, CFI = 0.969, RMSEA = 0.032, RMR = 0.023, SRMR = 0.032.

Even though the correlations obtained between factors were low, especially between factors 1 and 3 and between factors 2 and 3, a second-order model was carried out. The results indicated that the model did not suggest a second order factor, as factor 1 explains 111% of the global scale (Figure 4).

### 3.3. Convergent Validity and Composite Reliability 

Internal consistency of the scale ranged between low and barely acceptable depending on the results of the composite reliability for the different factors of the scale. On the other hand, AVE values suggested a lack of convergent validity as the indices found were higher than 0.50. The discriminant validity of the factors, assessed by comparing the AVE with the squared correlation, appeared only in factors 1 and 3 and between factors 2 and 3 (Table 4).

To conclude the validation study of the MHKQ, we calculated the internal consistency of the remaining items for each subscale. 

In Factor 1 (knowledge of the characteristics of mental health and mental disorders), Cronbach’s alpha coefficients indicated a barely acceptable internal consistency: the lowest value (α = 0.628) was found in item 16, and the highest value (α = 0.670) was reported in item 3, with an overall alpha of 0.693. In Factor 2 (belief in the epidemiology of mental disorders), Cronbach’s alpha coefficients presented a tendentially low internal consistency: the lowest value (α = 0.510) was obtained in item 14 and the highest (α = 0.536) in item 10, with an overall alpha of 0.597. In Factor 3 (awareness of mental health promotion activities), the internal consistency of Cronbach’s alpha coefficients was also tendentially low: indices ranged between 0.513 in item 19 and 0.558 in item 20, with an overall alpha of 0.602. Thus, Cronbach’s alpha threshold of 0.70 was not reached for any of the factors (Table 5).

Table 6 presents the convergent/divergent validity of the items. The presented outcomes highlight the existence of convergent validity between the different items, and the factor to which they correspond since the correlational value presented in bold is higher with the subscale to which it belongs and the second one with the total factor of the scale.

Finally, the correlation matrix between the three factors and the global scale is presented in Table 7. All correlations were positive and significant. The correlation between factor 1 and factor 3 (*r* = 0.345) is the highest and explains 11.90%. The variability found for the global scale is higher and ranges between 0.338—with factor 3 explaining 11.42%—and 0.817, with a variability of 66.74%. 

The statistics of the different factors were determined based on the sum of the items composing each factor. To compare the results obtained in the three subscales, since they are not composed of the same number of items, they were converted into the same level of magnitude on a scale ranging from 0 to 100, using the following formula: ((scale raw score-minimum score)/range) × 100. The final results are expressed in Table 8, with factors ranging between 0% and 100% and scores for the global scale ranging between 4.55% and 100%. The average indices showed that the lowest index was found in factor 3 (mean = 60.62 ± 30.97), while the highest average index was found in factor 1 (mean = 84.40 ± 13.14). The coefficients of variation indicated low dispersion for factor 1 and moderate dispersion for the remaining factors and the global scale. In turn, skewness and kurtosis values tended to show leptokurtic curves skewed to the right for factors 2 and 3 and skewed to the left for factor 1 and the global scale. 

Cut-off values were defined for each factor and the overall value of the sample based on the 25th and 75th percentiles, and the participants’ level of knowledge was classified as poor, moderate and good. Factor 3 was an exception since only poor and good levels of knowledge are reflected. As presented in Table 9, a good level of knowledge can be found in factor 1 (32.2%), while in factor 2 and the global scale, moderate knowledge obtained the highest score, while in factor 3, the percentage of respondents with poor and good knowledge is almost identical (50.5% and 49.5%, respectively).

## 4. Discussion

The translation and adaptation of the MHKQ to European Portuguese were based on (1) a technical review and semantic assessment adapted to a Likert-type scale; (2) content validation; and (3) administration of a pretest to a sample of the target population, to assess the participants’ degree of understanding. In the psychometric study of the scale, internal consistency was estimated using Cronbach’s alpha coefficient (α) and construct validity was assessed using exploratory and confirmatory factor analysis, convergent validity, and discriminant validity. Once the 20 statements were adapted to a 5-point Likert scale, exploratory factor analysis led to a three-factor structure that explained 40.86% of the total variance. Then, items 6 and 9 were removed since they showed loading levels lower than 0.40. Confirmatory factor analysis also led to the removal of items 1, 2, 8 and 15 since they all showed low loading levels. The overall goodness-of-fit indices were fairly adequate, confirming the quality of the model’s fit to the empirical data. The final Portuguese version of the MHKQ was composed of 14 items (Appendix A).

The final refined model with modification indices was obtained after correlated errors between item 4 (“All mental disorders are caused by external stressors”) and item 10 (“Even for severe mental disorders [e.g., schizophrenia], medications should be taken only for a given period”). The same method was applied to item 19 (“Have you heard about the International Day for Suicide Prevention?”) and item 20 (“Have you heard of World Sleep Day?”). We considered that the errors might be correlated for items 4 and 10 as they are both related to knowledge specifically focused on the causes and treatment of mental disorders, which are usually analysed comprehensively in studies about knowledge toward mental disorders and their treatment [25]. On the other hand, we considered that the errors might be correlated for items 19 and 20 because both concern to relatively recent mental health promotion days/activities, as they were celebrated for the first time in 2003 and 2008, respectively (while the International Mental Health Day and the International Day against Drug Abuse and Illicit Drug Trafficking are celebrated since 1992 and 1987, respectively).

Notably, we found low correlations between Factor 3 and the other factors in the final refined model with modification indices. That may be explained by two major reasons: (1) all items for Factor 3 were measured dichotomously, whereas the rest of the items (for Factors 1 and 2) were measured continuously; and (2) all items on Factor 3 relate to simple facts knowledge on mental health promotion days/activities, while the items on Factors 1 and 2 relate to respondents’ stereotypes, beliefs, and attributions.

Our findings also pointed out to a barely acceptable internal consistency of the MHKQ Factor 1, but low internal consistencies of Factors 2 and 3. This might be explained by the low number of questions of Factors 2 and 3 (four questions each) or the item content heterogeneity. However, this will need to be further explored.

It is interesting to compare and contrast the psychometrics of the MHKQ with the ones of other mental health literacy assessment tools, which have already been validated in Portugal. One of those tools is the Questionnaire for Assessment of Mental Health Literacy (QuALiSMental) [26]. That assessment tool presents acceptable levels of reliability and a factor structure consistent with the theoretical components of mental health literacy. However, this assessment tool is not easy to score, and it is mostly focused on knowledge about mental disorders.

The Mental Health Literacy Scale (MentaHLiS) [27] is a mental health literacy assessment tool aiming to assess mental health literacy in adolescents. The tool is composed of three scales. Each scale is divided into five subscales (recognition of the disorder; help-seeking recourses and options; help-seeking, perceived barriers and facilitators; first aid intentions and beliefs; and lifestyles and health behaviours). All subscales present acceptable reliability coefficients (0.705 to 0.811), better than the MHKQ. However, the tool also focusses on mental disorders, specifically depression, anxiety, and alcohol abuse.

The Mental Health Literacy Scale (MHLS) [14] presents a three-factor structure (attitudes towards mental illness; knowledge about mental illness; and ability to recognize symptoms), which explains 35% of the total variance (a value lower than the MHKQ). On the other hand, the internal reliability of the factors tends to be better than in the MHKQ (ranging from 0.66 and 0.89). However, as previously stated, this assessment tool tends to focus on the knowledge about mental disorders, such as social anxiety disorder, generalized anxiety disorder, or depression.

Finally, the Mental Health Literacy questionnaire (MHLq) was developed in 2016 and validated in a sample of adolescents [28]. Then, in 2018 [29], it was adapted for the young adult population. The final version of the questionnaire included 29 items, with good total internal consistency (α = 0.84), organized into four factors: knowledge of mental health problems; erroneous beliefs/stereotypes; help-seeking and first-aid skills; and self-help strategies. This assessment tool presents better psychometric properties than the MHKQ; however, it is a longer tool (29 items) compared to the MHKQ (14 items).

Thus, compared to the other mental health literacy assessment tools already validated in Portugal, the MHKQ presents acceptable psychometric properties, even though other tools present better psychometrics. Nonetheless, the MHKQ presents two important advantages: (1) it does not focus solely on the knowledge about mental disorders; and (2) it is a short (14 items) and easy to fill assessment tool.

This study enabled to translate, adapt and validate the MHKQ in a sample of higher education students in Portugal. This study has some limitations, even though the aim of the study was achieved. The most relevant of those limitations is that it was not possible to assess the external validity, measurement equivalence and reproducibility of the tool. Another potential limitation lies in the convenience sampling technique, as it limits the generalizability of results. Moreover, the MHKQ concurrent validity alongside an established mental health literacy instrument was not assessed, which can also be viewed as a limitation of this study. Finally, the low internal consistency of the MHKQ should also be pointed out as a limitation of the assessment tool itself.

## 5. Conclusions

The MHKQ was translated and adapted to European Portuguese in the form of a Likert-type scale. the produced results during content validation, verbal comprehension and internal consistency were rather positive. The factor model found in this study is somehow different from the theoretical proposal offered by the original version of the tool and the three-factor structure obtained in a study previously carried out in a sample of postpartum women. However, this version may still be considered valid and barely reliable for assessing students’ knowledge of mental health. Nevertheless, further research is still required to assess the external validity, measurement equivalence and reproducibility of the MHKQ.

## Figures and Tables

**Figure 1 ijerph-20-03022-f001:**
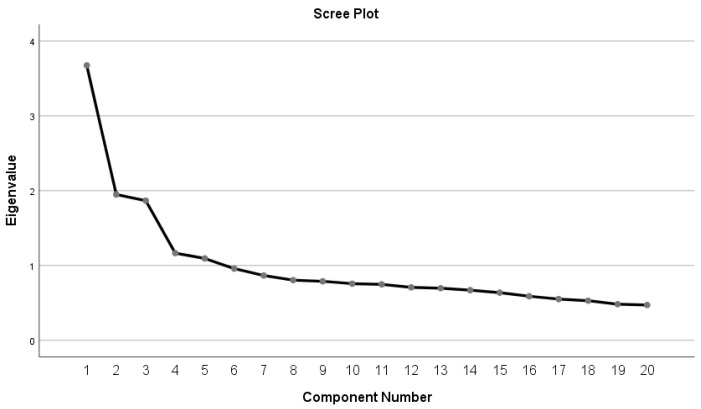
Scree plot.

**Figure 2 ijerph-20-03022-f002:**
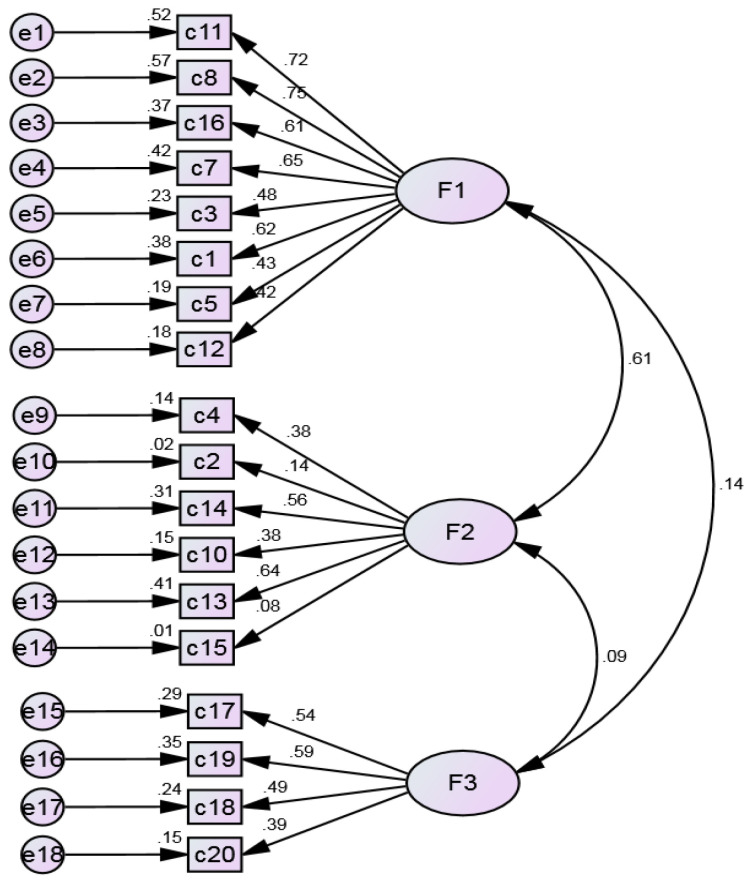
Early model.

**Figure 3 ijerph-20-03022-f003:**
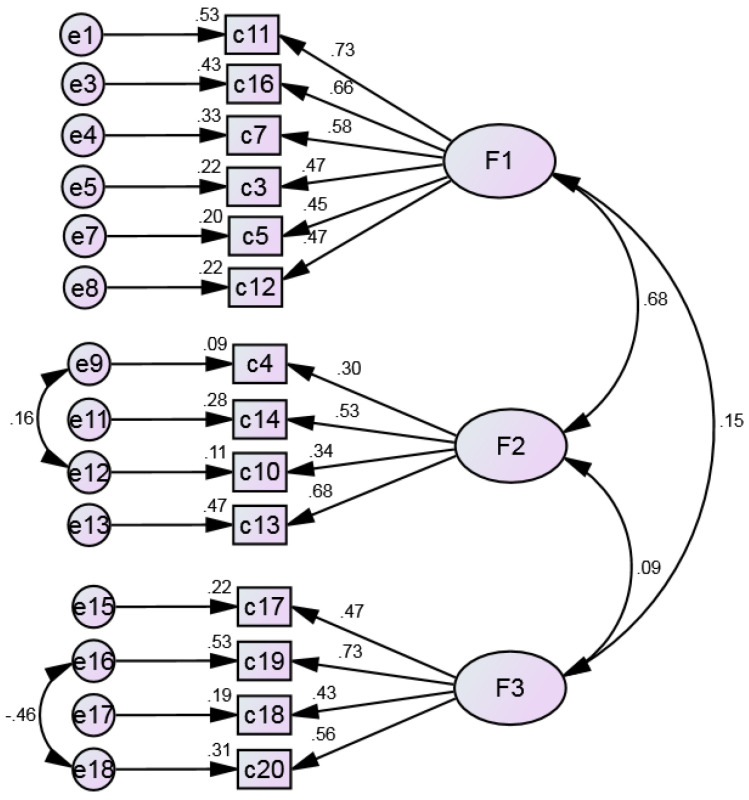
Final refined model with modification indices.

**Figure 4 ijerph-20-03022-f004:**
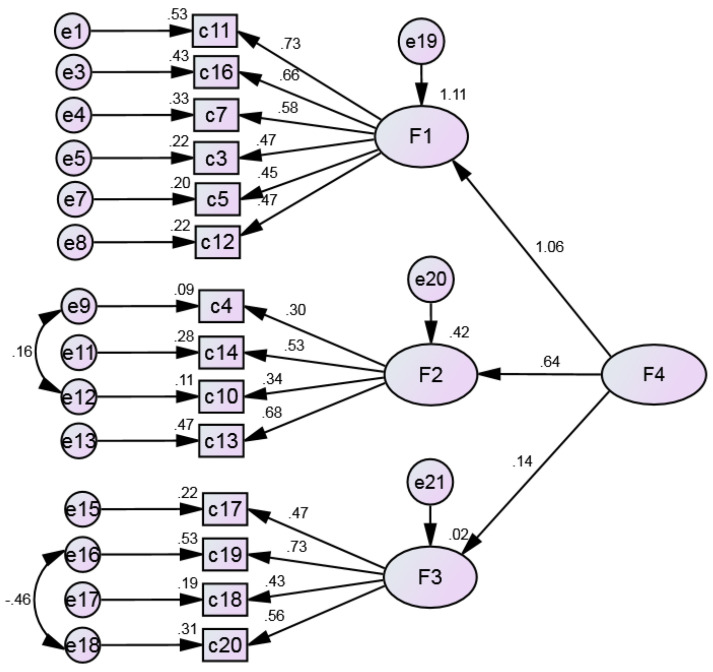
Second-order model.

**Table 1 ijerph-20-03022-t001:** Statistics (mean and standard deviation), item-total correlation, Cronbach’s alpha, and McDonalds’ omega coefficients of the Mental Health Knowledge Questionnaire.

	Mean	Standard Deviation	Item-Total Correlation	Cronbach’s α	McDonalds’ ω
1. Mental health is a component of health.	4.90	0.516	0.367	0.676	0.706
2. Mental disorders are caused by incorrect thinking.	2.96	1.218	0.174	0.694	0.728
3. Many people have mental problems but are not aware of it.	4.35	0.794	0.252	0.681	0.714
4. All mental disorders are caused by external stressors.	3.72	1.142	0.414	0.661	0.711
5. Mental health components include average intelligence, stable mood, a positive attitude, quality interpersonal relationship and adaptability.	4.14	0.945	0.250	0.681	0.714
6. Most mental disorders cannot be cured.	3.59	1.097	0.180	0.691	0.726
7. Psychological or psychiatric services should be sought if one suspects the presence of psychological problems or mental disorders.	4.65	0.721	0.371	0.671	0.704
8. Psychological problems can occur at almost any age.	4.78	0.600	0.440	0.669	0.697
9. Mental disorders and psychological problems cannot be prevented.	3.39	1.180	0.214	0.688	0.724
10. Even for severe mental disorders (e.g., schizophrenia), medications should be taken only for a given period; there is no need to take them for a long time.	3.92	1.152	0.391	0.664	0.710
11. Positive attitudes, good interpersonal relationships and a healthy lifestyle can help maintain mental health.	4.70	0.643	0.460	0.666	0.695
12. Individuals with a family history of mental disorders are at a higher risk for psychological problems and mental disorders.	4.04	0.934	0.270	0.679	0.714
13. Psychological problems in adolescents do not influence academic grades.	4.52	0.949	0.407	0.664	0.709
14. Middle-aged or elderly individuals are unlikely to develop psychological problems and mental disorders.	4.33	1.112	0.398	0.663	0.710
15. Individuals with bad temperament are more likely to have mental problems.	2.99	1.077	0.010	0.710	0.740
16. Mental problems or disorders may occur when an individual is under psychological stress and facing major life events (e.g., the death of family members).	4.37	0.837	0.392	0.667	0.702
17. Have you heard about International Mental Health Day?	0.81	0.389	0.185	0.687	0.722
18. Have you heard about International Day against Drug Abuse and Illicit Drug Trafficking?	0.37	0.483	0.013	0.696	0.733
19. Have you heard about International Suicide Prevention Day?	0.63	0.484	0.197	0.686	0.720
20. Have you heard about World Sleep Day?	0.61	0.487	0.083	0.692	0.729
Cronbach’s α	First half			0.563	
	Second half			0.469	
	Overall			0.691	
McDonalds’ ω	Overall			0.726	

**Table 2 ijerph-20-03022-t002:** Loadings of the items per factor, percentage of variance explained, and eigenvalues after rotation.

Items	Factor 1	Factor 2	Factor 3
1. Mental health is a component of health.	0.565	0.846	
2. Mental disorders are caused by incorrect thinking.		0.664	
3. Many people have mental problems but are not aware of it.	0.567		
4. All mental disorders are caused by external stressors.		0.700	
5. Mental health components include average intelligence, stable mood, a positive attitude, quality interpersonal relationships and adaptability.	0.539		
7. Psychological or psychiatric services should be sought if we suspect the presence of mental problems or disorders.	0.632		
8. Psychological problems can occur at almost any age.	0.682		
10. Even for severe mental disorders (e.g., schizophrenia), medications should be taken only for a given period.		0.542	
11. Positive attitudes, good interpersonal relationships and a healthy lifestyle can help maintain mental health.	0.711		
12. Individuals with a family history of mental disorders are at greater risk of psychological problems and mental disorders.	0.524		
13. Psychological problems in adolescents do not influence academic grades.		0.489	
14. Middle-aged or elderly individuals are unlikely to develop psychological problems and mental disorders.		0.555	
15. Individuals with bad temperament are more likely to have mental problems.		0.444	
16. Mental problems or disorders may occur when an individual is under psychological stress or facing a major life event (e.g., death of family members).	0.653		
17. Have you heard about International Mental Health Day?			0.697
18. Have you heard about the International Day against Drug Abuse and Illicit Drug Trafficking?			0.672
19. Have you heard about the International Day for Suicide Prevention?			0.657
20. Have you heard of World Sleep Day?			0.641
Percentage of variance explained	18.53%	11.79%	10.55%
Eigenvalues	3.33	2.12	1.90

**Table 3 ijerph-20-03022-t003:** Trajectories, critical ratios and lambda coefficients.

Trajectories			Estimates	SS	Critical Ratios	*p*	λ
c8	<---	F1	0.944	0.038	25.148	***	0.754
c16	<---	F1	0.969	0.046	21.123	***	0.609
c7	<---	F1	0.984	0.045	21.838	***	0.647
c3	<---	F1	0.786	0.047	16.573	***	0.482
c1	<---	F1	0.694	0.033	21.276	***	0.619
c5	<---	F1	0.825	0.055	15.067	***	0.432
c4	<---	F2	1.000				0.377
c2	<---	F2	0.395	0.094	4.211	***	0.142
c14	<---	F2	1.434	0.143	9.994	***	0.556
c10	<---	F2	1.004	0.112	8.964	***	0.382
c13	<---	F2	1.449	0.156	9.300	***	0.644
c15	<---	F2	0.196	0.082	2.387	0.017	0.080
c17	<---	F3	1.000				0.541
c19	<---	F3	1.314	0.133	9.874	***	0.592
c18	<---	F3	1.104	0.122	9.076	***	0.488
c20	<---	F3	0.892	0.092	9.740	***	0.391
c12	<---	F1	0.780	0.053	14.594	***	0.419
c11	<---	F1	1.000				0.724

*** *p* < 0.001.

**Table 4 ijerph-20-03022-t004:** Composite reliability, convergent validity and discriminant validity.

Factors	CR	AVE	Discriminant Validity		
			F1 vs. F2	F1 vs. F3	F2 vs. F3
Factor 1	0.733	0.322	0.464	0.022	
Factor 2	0.528	0.237			0.0009
Factor 3	0.636	0.314			
Global Scale	0.847	0.314			

**Table 5 ijerph-20-03022-t005:** Internal consistency of each factor of the Mental Health Knowledge Questionnaire.

Item		Mean	SD	r/Total Item	r^2^	α in Item	McDonalds’ ω
	Factor 1—Knowledge of the characteristics of mental health and mental disorders					0.693	0.709
3	Many people have mental problems but are not aware of it.	4.34	0.805	0.394	0.159	0.670	0.691
5	Mental health components include average intelligence, stable mood, a positive attitude, quality interpersonal relationships and adaptability.	4.15	0.951	0.390	0.161	0.669	0.684
7	Psychological or psychiatric services should be sought if we suspect the presence of mental problems or disorders.	4.63	0.742	0.445	0.227	0.655	0.672
11	Positive attitudes, good interpersonal relationships and a healthy lifestyle can help maintain mental health.	4.69	0.668	0.554	0.329	0.630	0.637
12	Individuals with a family history of mental disorders are at greater risk of psychological problems and mental disorders.	4.05	0.934	0.393	0.166	0.668	0.685
16	Mental problems or disorders may occur when an individual is under psychological stress or facing a major life event (e.g., death of family members).	4.40	0.817	0.508	0.278	0.628	0.649
	Factor 2—Belief in the epidemiology of mental disorders					0.597	0.601
4	All mental disorders are caused by external stressors.	3.71	1.149	0.338	0.116	0.534	0.547
10	Even for severe mental disorders (e.g., schizophrenia), medications should be taken only for a given period.	3.91	1.150	0.339	0.117	0.536	0.548
13	Psychological problems in adolescents do not influence academic grades.	4.50	0.966	0.367	0.155	0.527	0.528
14	Middle-aged or elderly individuals are unlikely to develop psychological problems and mental disorders.	4.32	1.119	0.390	0.172	0.510	0.513
	Factor 3—Awareness of mental health promotion activities					0.602	0.609
17	Have you heard about International Mental Health Day?	0.81	0.394	0.396	0.172	0.521	0.526
18	Have you heard about the International Day against Drug Abuse and Illicit Drug Trafficking?	0.38	0.485	0.368	0.144	0.534	0.557
19	Have you heard about the International Day for Suicide Prevention?	0.64	0.481	0.395	0.184	0.513	0.518
20	Have you heard of World Sleep Day?	0.60	0.489	0.320	0.116	0.558	0.572

**Table 6 ijerph-20-03022-t006:** Convergent/divergent validity of the items.

	Items	Factor 1	Factor 2	Factor 3	Total Scale
3	Many people have mental problems but are not aware of it.	0.598 ***	0.153 ***	0.061 **	0.458 ***
4	All mental disorders are caused by external stressors.	0.152 ***	0.663 ***	0.053 **	0.471 ***
5	Mental health components include average intelligence, stable mood, a positive attitude, quality interpersonal relationships and adaptability.	0.630 ***	0.134 ***	0.090 ***	0.474 ***
7	Psychological or psychiatric services should be sought if we suspect the presence of mental problems or disorders.	0.623 ***	0.239 ***	0.066 ***	0.522 ***
10	Even for severe mental disorders (e.g., schizophrenia), medications should be taken only for a given period.	0.210 ***	0.664 ***	0.128 ***	0.524 ***
11	Positive attitudes, good interpersonal relationships and a healthy lifestyle can help maintain mental health.	0.692 ***	0.301 ***	0.075 ***	0.600 ***
12	Individuals with a family history of mental disorders are at greater risk of psychological problems and mental disorders.	0.628 ***	0.225 ***	0.055 **	0.515 ***
13	Psychological problems in adolescents do not influence academic grades.	0.332 ***	0.636 ***	−0.005	0.550 ***
14	Middle-aged or elderly individuals are unlikely to develop psychological problems and mental disorders.	0.239 ***	0.691 ***	−0.009	0.524 ***
16	Mental problems or disorders may occur when an individual is under psychological stress or facing a major life event (e.g., death of family members).	0.688 ***	0.304 ***	0.073 ***	0.598 ***
17	Have you heard about International Mental Health Day?	0.109 ***	0.102 ***	0.647 ***	0.275 ***
18	Have you heard about the International Day against Drug Abuse and Illicit Drug Trafficking?	0.036	−0.052 *	0.681 ***	0.153 ***
19	Have you heard about the International Day for Suicide Prevention?	0.096 ***	0.111 ***	0.697 ***	0.283 ***
20	Have you heard of World Sleep Day?	0.058 **	0.029	0.651 ***	0.204 ***

* *p* < 0.05; ** *p* < 0.01; *** *p*< 0.001.

**Table 7 ijerph-20-03022-t007:** Pearson’s correlation matrix between the factors of the Mental Health Knowledge Questionnaire.

Factors	Factor 1	Factor 2	Factor 3	Global Scale
Factor 1	1			
Factor 2	0.345 ***	1		
Factor 3	0.109 ***	0.067 ***	1	
Global Scale	0.817 ***	0.777 ***	0.338 ***	1

*** *p* < 0.001.

**Table 8 ijerph-20-03022-t008:** Statistics concerning the Mental Health Knowledge Questionnaire.

	Min	Max	Mean	SD	CV (%)	Sk/Error	K/Error
Factor 1	0	100	84.40	13.14	15.56	40.80	74.78
Factor 2	0	100	77.72	18.20	23.61	−24.00	18.76
Factor 3	0	100	60.62	30.97	51.08	−8.021	8.68
Global Scale	4.55	100.0	79.81	11.94	38.55	25.47	32.69

**Table 9 ijerph-20-03022-t009:** Classification of the respondents’ level of knowledge for each subscale.

Knowledge	Poor		Moderate		Good	
Factors	*n*	%	*n*	%	*n*	%
Factor 1	928	32.1	826	28.6	1133	39.2
Factor 2	907	31.4	1196	41.4	784	27.2
Factor 3	1457	50.5			1430	49.5
Global Scale	753	26.1	1326	45.9	808	28.0

## Data Availability

Not applicable.

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
