# Peer review of "Translation, Adaptation and Assessment of the Psychometric Properties of the Mental Health Knowledge Questionnaire in a Sample of Higher Education Students in Portugal"

_ijerph, 2023, doi:10.3390/ijerph20043022_

Round 1

Reviewer 1 Report

Thank you for the opportunity to review your well-written manuscript. I hope my suggestions below are helpful:

Abstract

6th sentence: Please capitalize the “i” in “internal consistency for the psychometric study…”

7th sentence: I would argue that construct validity is demonstrated or tested rather than performed. Please consider using those words instead.

11th sentence: I would recommend replacing the word “construct” with “scale” (e.g., “Analyses to confirm the scale’s external validity…”). As I understand it, latent constructs themselves cannot have validity but rather validity is a property of the scales we have created to measure the constructs.

Introduction

p.2, line 51 - I would consider replacing the word “activist” with “proactive”

Materials and Methods

p.2, line 81: I would rewrite the first sentence to state: “A psychometric study was carried out since this is the most commonly used approach to validate self-report instruments intended to measure latent constructs.”

Extremely clear and stepwise accounting of data analysis procedures - bravo!

Results

p.7, line 305 - Please recheck. My understanding is that p = 0.001 for Bartlett’s Test of Sphericity indicates that the items are in fact correlated - if there were no intercorrelations among the items, then EFA would not be appropriate.

p. 7, line 306 - I would recommend moving away from the language of “proves.” Instead, consider writing something like: “These test results support the use of exploratory factor analysis with these data.”

Discussion

Please consider adding 1-2 sentences about the correlated errors in the final refined model with modification indices - do the authors have any hypotheses about why these errors might be correlated for these items?

Please consider discussing the low correlations between Factor 3 and the other factors in the final refined model with modification indices - conceptually or theoretically, how do the items that make up this hypothesized facet of mental health knowledge differ from the others? (e.g., all items for this factor were measured dichotomously whereas the rest of the items are measured continuously, or all items on this factor relate to simple declaration of fact versus respondents’ stereotypes, beliefs, and attributions; etc).

Author Response

Dear reviewer,

We would like to thank you very much for your recommendations, which we found quite valuable in order to improve the overall quality of the paper.

Reviewer 2 Report

I think the manuscript deals with a really important topic, as it tries to adapt an instrument that aims to assess the population's knowledge and awareness of mental health. I think that instrument adaptation and validation is a really important topic and it seems to me that this is an area where not much is published because journals in general do not facilitate the publication of articles in this area. Therefore, I think the publication of this manuscript would be really interesting. However, there are certain issues that I think could be improved before publication.

-          Incorporate a quotation in the statement:  “The literature also argues that investigating the levels of mental health literacy among higher education students will contribute to fighting the stigma towards 57 mental disorders, demystifying false beliefs and stereotypes, and providing a new inclusive approach to every member of society.” Page 2, lines 56-56

-          Further justification of the MHKQ scale as a tool for assessing the positive component of mental health literacy.

Author Response

(The authors gave the same response as above.)
